# Multitask Learning Can Improve Worst-Group Outcomes

**Atharva Kulkarni**[*]                                               *atharvak@cs.cmu.edu*
*Language Technologies Institute, School of Computer Science*
*Carnegie Mellon University*

**Lucio M. Dery**[*]                                                  *ldery@cs.cmu.edu*
*Computer Science Department, School of Computer Science*
*Carnegie Mellon University*

**Amrith Setlur**                                                     *asetlur@cs.cmu.edu*
*Machine Learning Department, School of Computer Science*
*Carnegie Mellon University*

**Aditi Raghunathan**                                                 *raditi@cs.cmu.edu*
*Computer Science Department, School of Computer Science*
*Carnegie Mellon University*

**Ameet Talwalkar**                                                   *atalwalk@cs.cmu.edu*
*Machine Learning Department, School of Computer Science*
*Carnegie Mellon University*

**Graham Neubig**                                                     *gneubig@cs.cmu.edu*
*Language Technologies Institute, School of Computer Science*
*Carnegie Mellon University*

**Reviewed on OpenReview:** *https://openreview.net/forum?id=sPlhAIp6mk*

## Abstract

In order to create machine learning systems that serve a variety of users well, it is vital to not only achieve high average performance but also ensure equitable outcomes across diverse groups. However, most machine learning methods are designed to improve a model's average performance on a chosen end task without consideration for their impact on worst group error. Multitask learning (MTL) is one such widely used technique. In this paper, we seek not only to understand the impact of MTL on worst-group accuracy but also to explore its potential as a tool to address the challenge of group-wise fairness. We primarily consider the standard setting of fine-tuning a pre-trained model, where, following recent work (Gururangan et al., 2020; Dery et al., 2023), we multitask the end task with the pre-training objective constructed from the end task data itself. In settings with few or no group annotations, we find that multitasking often, but not consistently, achieves better worst-group accuracy than Just-Train-Twice (JTT; Liu et al. (2021)) – a representative distributionally robust optimization (DRO) method. Leveraging insights from synthetic data experiments, we propose to modify standard MTL by regularizing the joint multitask representation space. We run a large number of fine-tuning experiments across computer vision and natural language processing datasets and find that our regularized MTL approach *consistently* outperforms JTT on both average and worst-group outcomes. Our official code can be found here: **https://github.com/atharvajk98/MTL-group-robustness**.

---

[*]Equal contribution.

# 1 Introduction

As machine learning systems exert ever-increasing influence on the real world, it is paramount that they not only perform well on aggregate but also exhibit equitable outcomes across diverse subgroups characterized by attributes like race (Buolamwini & Gebru, 2018; Liang et al., 2021), gender (Buolamwini & Gebru, 2018; Srinivasan & Bisk, 2022) and geographic location (Jurgens et al., 2017; De Vries et al., 2019; Ayush et al., 2021). Therefore, it is important to understand the impact of widely-used machine learning techniques with respect to these desiderata. Multitask learning (MTL) (Caruana, 1997; Baxter, 2000; Ruder et al., 2019; Dery et al., 2021a) is one example of such a technique that features prominently in machine learning practitioners' toolbox for improving a model's aggregate performance. However, the effect of multitask learning on worst group outcomes is underexplored. In this paper, we both study the impact of MTL, *as is*, on worst group error and also consider whether modifications can be made to improve its effect on worst group outcomes.

Traditionally, the problem of worst-group generalization has been tackled explicitly via methods such as distributionally robust optimization (DRO) (Ben-Tal et al., 2013; Duchi & Namkoong, 2018; Hashimoto et al., 2018; Sagawa et al., 2020a). In contrast to traditional empirical risk minimization, DRO aims to minimize the worst-case risk over a predefined set of distributions (the *uncertainty set*). Defining the uncertainty set usually (but not always) requires access to group annotations. Since average performance based approaches like MTL are typically designed without consideration of group annotations, our focus will be on settings with limited-to-no group annotations. In these settings, there exist a number of *generalized Reweighting (GRW)* algorithms for distributional robustness (Nam et al., 2020; Liu et al., 2021; Zhang et al., 2022; Nam et al., 2022; Qiu et al., 2023; Zhai et al., 2023; Izmailov et al., 2022). These approaches minimize the weighted average risk based on the weight assigned to each example. One such widely used method is the Just-Train-Twice (JTT) algorithm (Liu et al., 2021), which performs two model training runs: one to identify poorly performing examples and another run that upweights these examples. For our empirical explorations, we take JTT as a representative DRO method and use it to provide reference performance to situate our study of multitask learning.

We focus our investigations of multitask learning on the ubiquitous setting of fine-tuning a pre-trained model. Here, a common way to improve end task average performance is to multitask the end task with the pre-training objective constructed over the task data itself (Gururangan et al., 2020; Dery et al., 2021b). We intuit that this multitasking approach could improve robustness to worst group outcomes since previous work like Hendrycks et al. (2019; 2020); Mao et al. (2020) has established a favorable connection between pre-training and robustness (both adversarial and out-of-distribution). We test our intuition by conducting

Table 1: Standard multitasking improves worst group outcomes over ERM and JTT **but not consistently**. Experimental details can be found in Section A.1.

| Dataset | Method | No Group Labels | Val Group Labels |
|---|---|---|---|
| | | Worst-Group Acc | Worst-Group Acc |
| Waterbirds | ERM | $80.1_{4.6}$ | $85.4_{1.4}$ |
| | JTT | $\mathbf{82.1}_{1.2}$ | $\mathbf{85.9}_{2.5}$ |
| | (MTL) ERM+MIM | $80.1_{4.6}$ | $85.3_{2.4}$ |
| Civil-Small | ERM | $51.6_{5.6}$ | $67.4_{2.1}$ |
| | JTT | $52.5_{5.2}$ | $68.0_{1.8}$ |
| | (MTL) ERM+MLM | $\mathbf{58.3}_{6.6}$ | $\mathbf{68.5}_{0.4}$ |

preliminary experiments across a pair of computer vision and natural language tasks in two settings: one with limited group annotations and the other with none. Initial results (Table 1) reveal that multitasking shows promise in that it can improve worst group outcomes over ERM and JTT/ However, these improvements are not consistent. Therefore, we are spurred to consider modifications to make it a more competitive tool against worst group outcomes.

In order to build intuition about how to adapt MTL to target the worst group error, we conduct controlled experiments on two-layer linear models trained from scratch on synthetic data. We borrow the synthetic data setup introduced by Sagawa et al. (2020b) in which training data consists of two majority groups, where spurious features (features that are not required to robustly solve the end task) are predictive of the end task output, and two minority groups, where spurious features are uncorrelated with the output. Sagawa et al. (2020b) demonstrated that under certain conditions on the generative distribution of the input features, linear models trained on such data would probably rely on spurious features and thus, have poor worst group error. Working with this simplified setup where worst-group outcomes are easily inducible allows us to study MTL's effects more incisively. We instantiate reconstruction from noised input as our auxiliary

task to perform multitask learning in the synthetic setup. This choice is partially[*] informed by the fact that many pre-training objectives like masked language modeling (MLM) (Devlin et al., 2018) and masked image modeling (He et al., 2022; Tong et al., 2022) are based on input reconstruction. When training solely on this auxiliary task, we uncover that regularizing the pre-output layer of the model is critical for ensuring that the model upweights the core features (features required to robustly solve the end task) over the spurious ones. This leads us to the following recipe for improving worst-group error: regularized multitask learning of the end task with the (appropriately chosen) auxiliary objective.

Through a battery of experiments across natural language processing (NLP) and computer vision (CV) datasets, we demonstrate that multitasking the end task with the pre-training objective along with $\ell_1$ regularization on the shared, pre-prediction layer activations is competitive when pitted against state-of-the-art DRO approaches like JTT (Liu et al., 2021) and Bitrate-Constrained DRO (Setlur et al., 2023). Specifically, in settings where only validation group annotations are available, regularized MTL outperforms JTT and BR-DRO on 3/3 and 2/3 datasets, respectively. Our approach improves worst-group accuracy over ERM (by as much as $\sim 4\%$) and JTT (by $\sim 1\%$) in settings where group annotations are completely unavailable. Moreover, regularized MTL consistently outperforms both ERM and JTT on average performance, regardless of whether group annotations are available or not. Thus, within the prevailing framework of utilizing pre-trained models for downstream fine-tuning, our results demonstrate that regularized multitask learning can be a simple yet versatile and robust tool for improving both average and worst-group outcomes.

## 2 Informal motivation for our regularized MTL method

**Problem Setup / Preliminaries:** Let each input example $x \in X$ have a classification label $y \in Y$ and a demographic attribute $s \in S$. Each group $g = (s, y) \in G$ is defined by the label $y$ and the attribute $s$, such that $s$ spuriously correlates with $y$. Thus, the sample space of $G$ is the Cartesian product of $Y$ and $S$. Our goal is to learn a model that minimizes the worst-group error. We evaluate the models based on their worst group accuracy (WGA), i.e., the minimum predictive accuracies of our model across all groups. We are interested in the setting where the spurious attribute $s$, and consequently, the group identity $g$ are unavailable (or available to a very limited degree) at training time, as annotating spurious attributes is typically expensive.

**Why would we expect multitask learning to help mitigate worst group outcomes?** It would be naive to assume that multitasking the end task with *any* auxiliary task would prevent poor group outcomes. In order to better understand intuitively which auxiliary tasks may be helpful, we first provide an example using the data generation process and linear model setup presented in Sagawa et al. (2020b)'s work on the effect of spurious and core features on worst-group accuracy.

**When do models incur high worst group error?** Sagawa et al. (2020b) describe a simple data-generating distribution that defines, for each example, a label $y \in \{-1, 1\}$, a spurious attribute $s \in \{-1, 1\}$, and features $x$. The features are described as either core features $x_{\text{core}}$ if they are associated with the label $y$, or spurious features $x_{\text{spur}}$ if they are associated with the spurious attribute $s$:

$$
\begin{aligned}
x_{\text{core}} \mid y &\sim \mathcal{N}\left(y\mathbf{1} \, , \, \sigma_{\text{core}}^2 I_{d_c}\right) \\
x_{\text{spur}} \mid s &\sim \mathcal{N}\left(s\mathbf{1} \, , \, \sigma_{\text{spur}}^2 I_{d_s}\right)
\end{aligned}
\tag{1}
$$

$$
x = [x_{\text{core}}; x_{\text{spur}}] \in \mathbb{R}^d \quad \text{and} \quad d = (d_c + d_s)
$$

We can then define a linear model, parameterized by $\hat{\mathbf{w}}$, that predicts the label given the features.

$$
\hat{y}^{(i)} = \hat{\mathbf{w}} \cdot x^{(i)}.
\tag{2}
$$

Note that because the core features $x_{\text{core}}$ are the ones associated with the label to be predicted, they are the ones that the model *should* use in order to attain high predictive accuracy.

---

[*]We will delve deeper into other motivations in Section 2

The cross-product of the space of possible labels $y = \pm 1$ and spurious attributes $s = \pm 1$ divides the samples generated from this distribution into four *groups*. When some groups are more frequent than others in the training data, a correlation between $\{y, s\}$ is created. Further still, in the presence of the above correlation, if the spurious features have lower variance with respect to the data generating process (Equation 1), i.e., $\sigma_{\text{spur}}^2 \leq \sigma_{\text{core}}^2$, linear models will tend to rely more on (assign a higher weight to) the spurious features over the core ones (Sagawa et al., 2020b). This learned reliance on the spurious features – instead of the core features that are truly predictive of the label – results in poor worst-group error.

**Why does reconstruction help?**   Considering the above, for an auxiliary task to be helpful, it should discourage the model from using spurious features by showing a stronger preference for core features in exactly the case when $\sigma_{\text{spur}}^2 \leq \sigma_{\text{core}}^2$. In this paper, we argue that one class of tasks that fulfills this criterion are *reconstruction tasks*, where we predict original input features from noised versions.

For instance, in the example above, if we add noise with a constant variance of $\sigma_{\text{noise}}^2$ over each dimension, it results in noised inputs that have variances $\tilde{\sigma}_{\text{spur}}^2 = \left(\sigma_{\text{spur}}^2 + \sigma_{\text{noise}}^2\right) \leq \tilde{\sigma}_{\text{core}}^2 = \left(\sigma_{\text{core}}^2 + \sigma_{\text{noise}}^2\right)$ per spurious and core feature dimension, respectively. Under the assumption that both true labels $y = 1$ and $y = -1$ are equally probable, and in the simplest case where we are reconstructing features independently of each other with a linear predictor $\hat{x}_i = \boldsymbol{w}_i \tilde{x}_i$ (where $\tilde{x}$ is the noised input), the Bayes optimal weight on a feature $i$ would be (see Appendix A.3 for the full proof):

$$\boldsymbol{w}_i^{\text{bayes}} = \frac{\sigma_i^2 + 0.5\left(\mu_{i|y=1}^2 + \mu_{i|y=-1}^2\right)}{\sigma_i^2 + 0.5\left(\mu_{i|y=1}^2 + \mu_{i|y=-1}^2\right) + \sigma_{\text{noise}}^2} \tag{3}$$

where $\mu_{i|y=\pm 1}^2$ are the per-dimension means from Eqn 1

Note that $\boldsymbol{w}_i^{\text{bayes}}$ is larger for dimensions with higher variances $\sigma_i^2$, assuming $\mu_{i|y=\pm 1}^2$ are symmetric across core and spurious features (i.e all $i$). Thus, this reconstruction task places more weight on the core features in exactly the setting where a linear predictor for the end task would prefer to use the spurious features. Note that for this preference of the core features to be effectively realized, the auxiliary task needs to be sufficiently up-weighted, but not so much so that the end task is not learned at all.

**Why is regularization necessary?**   Given an auxiliary task with the above property of preferring core to spurious features (under $\sigma_{\text{spur}}^2 \leq \sigma_{\text{core}}^2$), a model with sufficient capacity can still rely on spurious features for solving the end task. We can incentivize the model to mostly use core features by applying sufficient regularization to the parts of the model that are shared between the two tasks (such as shared feature extractors). The restricted capacity encourages the model to rely on features that would cause it to do well on **both** tasks, which would be the core features.

Based on the intuition established in this section, we propose a simple yet effective method for improving worst-group outcomes: *Multitasking the end task with the pre-training objective – which tends to be reconstruction tasks – while regularizing the shared (pre-prediction) layer*. In Section 3, we will test this intuition through synthetic data experiments, and in Sections 4 and 5, we demonstrate its empirical efficacy through natural data experiments.

## 3   Synthetic Data Experiments

We initiate our investigation with an empirical study in a simplified context, involving the training of a two-layer linear model on synthetic data. This exploration is designed to substantiate the informal intuition introduced in Section 2 through concrete empirical findings.

### 3.1   Data Generating Distribution

We base our experiment on the data generation distribution from Equation 1, where features are divided into core and spurious ones.

As an instantiation, we consider end the task data defined by $\mathbf{T}_{\text{end}} = \{(x_i, y_i)\}_{i \in N}$ where we have $N$ total samples. Here $d_c = 1; \; d_s = 1 \implies d = 2$. The data is dominated by samples where $\{s = y\}$ and thus, we have two majority groups $\mathbf{G}_{s=y=1}$ and $\mathbf{G}_{s=y=-1}$, each with $\frac{n_{\text{maj}}}{2}$ samples. The two minority groups are when $\{s = -y\}$ each with $\frac{n_{\text{min}}}{2}$: $\mathbf{G}_{s=-y=1}, \mathbf{G}_{s=-y=-1}$ . Due to the fact that $n_{\text{maj}} > n_{\text{min}}$, the attribute $s$ is highly correlated with the label $y$ in the training data and thus, is a spurious feature when considering the true data generation distribution. The end task is to predict the true label $y_i$ from the given input data $x_i$.

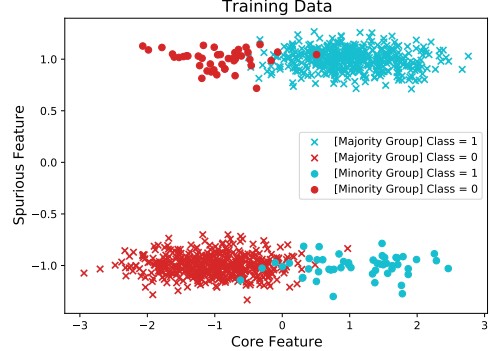

Figure 1: Visualization of synthetic training data (1000 points).

Figure 1 shows data sampled from the generative process described in Equation 1. We produce 1000 samples in $\mathbf{R}^2$ with $\sigma_{\text{core}}^2 = 0.6$ and $\sigma_{\text{spur}}^2 = 0.1$. $n_{\text{min}} = 100$ and $n_{\text{max}} = 900$, making the spurious feature highly correlated with the true label.

## 3.2 Training on the end task only

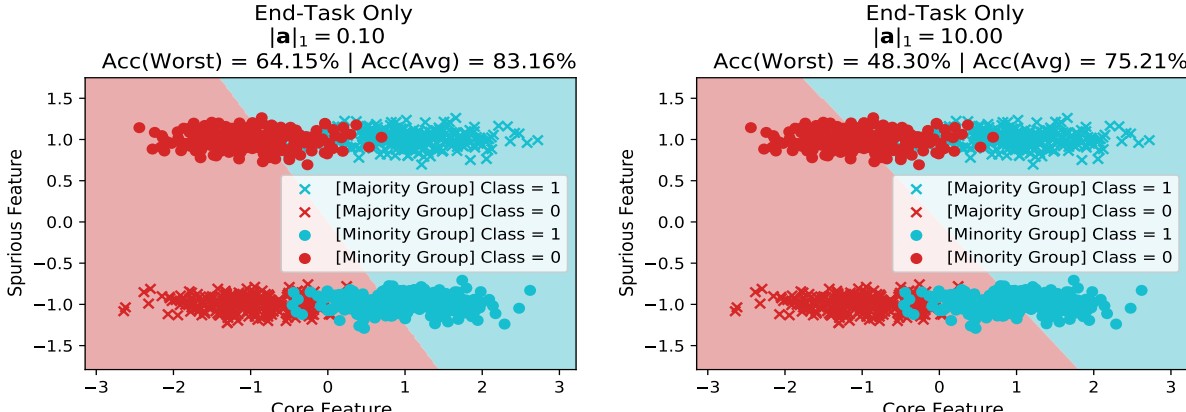

Figure 2: Predictors learned when we train on the end task only. Examples visualized are the balanced test samples created from Equation 1.

We train on $\mathbf{T}_{\text{end}}$ only to confirm that the resulting model has poor worst-group outcomes. Since we will eventually be do multitasking, we use a two layer linear model where the first layer is a linear featurizer – that will eventually be shared between all tasks being multitasked – and the second layer is a prediction head dedicated to the end task. This shared featurizer but separate head architecture is common in modern multitask learning (Yu et al., 2020; Michel et al., 2021; Dery et al., 2021a).

For simplicity, the featurizer layer $f(\cdot)$ is a diagonal linear function parameterized by $\mathbf{a}^*$:

$$f : \mathbb{R}^d \to \mathbb{R}^d \mid f_{(a)}(x) = (\mathbf{diag}(\mathbf{a})) \, x \tag{4}$$

And the final output prediction layer is given by

$$y_{\text{pred}}^{\text{end}} = \left(w^{\text{end}}\right)^T f(x) = (w^{\text{end}})^T \left(\mathbf{diag}(\mathbf{a})\right) x = \left(\hat{w}^{\text{end}}\right)^T x$$

Note that we have effectively parameterized a linear model with decomposed formulation which will be helpful once we proceed to multitasking. The end task loss is binary cross-entropy with $\ell_2$ regularization on

---

*the diagonal parameterization allows us to easily read off how much weight is assigned to core features versus spurious ones

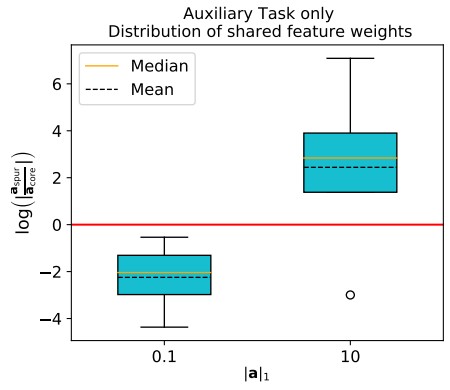

Figure 3: The ratio $\log(\frac{\mathbf{a}_{\text{spur}}}{\mathbf{a}_{\text{core}}})$ for 2 (extreme) choices of $|\mathbf{a}|_1$ across 4 hyperparameter settings (learning rate × batch size).

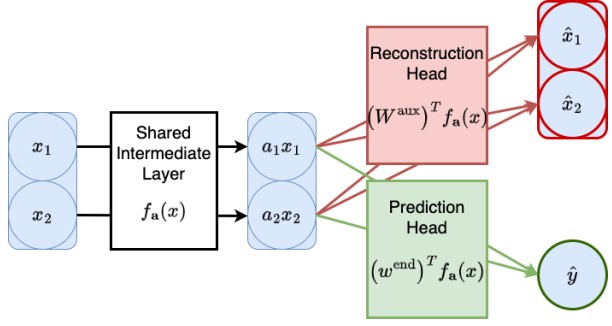

Figure 4: Multitask learning architecture used in Section 3.4. We use a shared intermediate layer and two separate prediction heads for $\mathbf{T}_{\text{aux}}$ and $\mathbf{T}_{\text{end}}$

$w^{\text{end}}$ is given by the following equation where $\sigma$ be the sigmoid function:

$$\mathcal{L}_{\text{end}}\left(w^{\text{end}}, a\right) = \frac{1}{N} \sum_{(x_i, y_i) \in \mathbf{T}_{\text{end}}} \left[ y_i \cdot \log\left(\sigma\left(y_{\text{pred}}^{\text{end}}\right)\right) + (1 - y_i) \cdot \log\left(1 - \sigma\left(y_{\text{pred}}^{\text{end}}\right)\right) \right] + \frac{\lambda}{2}\|w^{\text{end}}\|^2 \quad (5)$$

We fit the model solely to the end task by running batched stochastic gradient descent on $\mathcal{L}_{\text{end}}$. We use a batch size of 64, learning rate of $10^{-3}$, $\lambda = 1$, and 500 epochs. We use 100 generated points as validation data for model selection. As can be seen in Figure 2, training on the end task only can result in a predictor that achieves poor worst group error. This occurs even with varying the norm of the featurizer parameter $\mathbf{a}$.

### 3.3 Training on auxiliary data only

As motivated in Section 2, we proceed to introduce a reconstruction based auxiliary task. The auxiliary task data is defined by $\mathbf{T}_{\text{aux}} = \{(\tilde{x}_i, x_i)\}_{i \in M}$ where we have $M$ total samples. $M$ unlabelled points (with respect to the end task) are taken from the distribution described by Equation 1. Noise of the form

$$\epsilon_{\text{noise}} \sim \mathcal{N}\left(0, \sigma_{\text{noise}}^2 I_d\right) \quad | \quad \tilde{x} = x + \varepsilon \quad (6)$$

is applied to each point. The task is to reconstruct $x_i$ from $\tilde{x}_i$. Reusing the featurizer from Equation 4, we define the following prediction model:

$$x_{\text{pred}}^{\text{aux}} = \left(W^{\text{aux}}\right)^T f_{(\mathbf{a})}(\tilde{x})$$

$W^{\text{aux}}$ parameterizes the auxiliary prediction head which we regularize to $\{W^{\text{aux}} \in \mathbb{R}^{d \times d} \mid \|W^{\text{aux}}\|_F^2 = 1\}$. Finally, our reconstruction loss is given by:

$$\mathcal{L}_{\text{recon}}\left(W^{\text{aux}}, a\right) = \frac{1}{2M} \sum_{(\tilde{x}_i, x_i) \in \mathbf{T}_{\text{aux}}} \|x_i - \left(W^{\text{aux}}\right)^T f(\tilde{x})\|^2 \quad (7)$$

Using the synthetic data instantiation in Figure 1, we apply noise from $\mathcal{N}\left(\mathbf{0}, I_2\right)$, i.e., $\sigma_{\text{noise}}^2 = 1$ on each of the 1000 training points to get the training data for the auxiliary task [*]. We fit the model solely to the auxiliary task by running batched stochastic gradient descent on $\mathcal{L}_{\text{recon}}$. We use learning rates in the set $\{10^{-2}, 10^{-3}\}$ and batch sizes in the set $\{64, 256\}$.

We consider two cases when the intermediate layer $\mathbf{a}$ has low versus high capacity, as reflected in $\ell_1$-norm. Low $\ell_1$-norm $- \|\mathbf{a}\|_1 = |\mathbf{a}_{\text{spur}}| + |\mathbf{a}_{\text{core}}| = 0.1$ means restricted capacity since this constraint (along with $\|W^{\text{aux}}\|_F = 1$) results in models that cannot fit the training data perfectly. High $\ell_1$-norm ($\|\mathbf{a}\|_1 = 10$)

---

[*]Note that while we could generate more points for the auxiliary task, we would like to mimic the setting where we refrain from introducing external data (data beyond end task training data) since methods like JTT and BR-DRO do not utilize them.

means that the model is expressive enough to perfectly fit the training data. Figure 3 provides insight into the learned intermediate layer in either case. When the model has enough capacity, there is no competition between the core and spurious features. This means solutions where the spurious feature is weighted more than the core feature are feasible as long as the core feature weight is enough to reconstruct the the noised core features well.

However, under restricted capacity where the learned weight of the core and spurious features are in direct competition, the model has to put more weight on the core features in order to achieve a lower auxiliary loss (as motivated in Section 2). This can be seen in Figure 3. Thus, for the auxiliary task to be effective at forcing a model to use core features over spurious ones, the model's capacity must be reasonably restricted.

### 3.4 Multitasking with regularization

Given the findings from Section 3.3, we proceed to multitask $\mathcal{L}_{end}$ and $\mathcal{L}_{recon}$ along with regularization on the shared layer $\mathbf{a}$. Let $\mathbf{A}\left(\tau\right) = \{\mathbf{a} \in R^d \mid |\mathbf{a}|_1 = \tau\}$ be a set of $\ell_1$-norm constrained vectors, we solve the following multitask optimization problem:

$$\tilde{\mathbf{W}}^{aux}, \tilde{\mathbf{w}}^{end}, \tilde{\mathbf{a}} = \operatorname{argmin}_{\|W^{aux}\|_F^2=1, \ \mathbf{a} \ \in \ \mathbf{A}(\tau)} \quad \alpha \cdot \mathcal{L}_{recon}\left(W^{aux}, \mathbf{a}\right) + \mathcal{L}_{end}\left(w^{end}, \mathbf{a}\right) \tag{8}$$

We implement the multitask model illustrated in Figure 4. We use the same set of hyper-parameters as used in Section 3.3 and perform joint stochastic gradient descent on both $\mathbf{T}_{end}$ and $\mathbf{T}_{aux}$. When $\tau$ is chosen to be small enough, model capacity is restricted, and the model is forced to rely chiefly on the core features to do well on both the end and auxiliary tasks. Figure 5 evinces this. When the norm of the shared layer is high $\|\mathbf{a}\|_1 = 10$, the end task can still predominantly rely on the spurious features leading to poor

Table 2: Summary of results from Sections 3.3 and 3.4. Regularized multitasking leads to improved worst-group outcomes.

| Method | $\|\mathbf{a}\|_1 = 0.1$ | $\|\mathbf{a}\|_1 = 10$ |
|---|---|---|
| | Worst-Group Acc | Worst-Group Acc |
| End task only | 64.15 | 48.30 |
| Regularized MTL | **94.02** | 0.0 |

worst group error. On the other hand, when model capacity is reasonably restricted by setting $\|\mathbf{a}\|_1 = 0.1$, we see from Figure 5 (left) that we can achieve improved worst group accuracy. Thus, we can effectively leverage the reconstruction auxiliary task in this simplified setting by applying sufficient regularization to ensure improved worst-group outcomes.

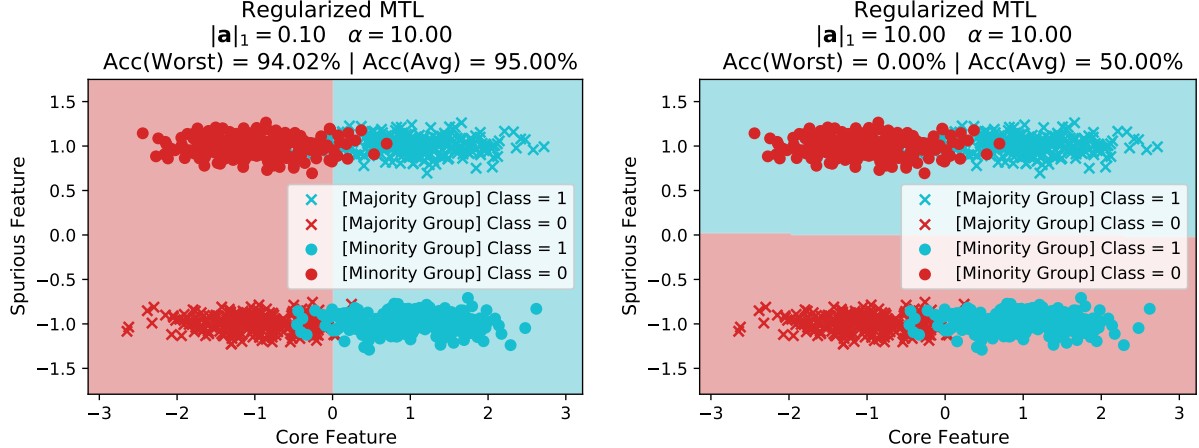

Figure 5: Depicted are the learned half-spaces for the multitask model under $\tau = \{0.1, 10\}$ and $\alpha = 10$. Restricting the capacity of the shared feature space is critical for multitasking to be effective for improving worst group error. Examples visualized are 1000 balanced test examples sampled from Equation 1

# 4 Details For Natural Data Experiments

We have made a case for using regularized multitask learning to combat poor worst-group performance through empirical explorations in a simplified, synthetic setting. In this section, we review the experimental details for our investigations of tasks of more practical interest.

## 4.1 Datasets

We conduct experiments across three datasets. To relieve the burden of compute, we introduce a fourth dataset, a smaller, sub-sampled version of one of the original datasets for ablations.

1. **Waterbirds:** This image classification dataset was introduced by Sagawa et al. (2020a). The task is to distinguish between species of land and water birds. It consists of bird images sourced from the CUB dataset Wah et al. (2011) and superimposed on land or water backgrounds from the Places dataset Zhou et al. (2018). The label (type of bird) is spuriously correlated with the background, resulting in 4 groups. Since this is a small dataset (4795 train examples), we also use it for ablations.
2. **MultiNLI:** This is a natural language inference dataset. The task is to classify whether the second sentence is entailed by, contradicts, or is neutral with respect to the first sentence (Williams et al., 2018). Following Sagawa et al. (2020a), we utilize the presence of negation words as a spurious attribute, leading to the creation of a total of 6 groups.
3. **Civilcomments:** The Civilcomments dataset is a toxicity classification dataset that contains comments from online forums Borkan et al. (2019); Koh et al. (2021). Along with the toxicity label, each text is annotated with additional overlapping sub-group labels of 8 demographic identities: male, female, LGBTQ, Christian, Muslim, other religions, Black, and White. As per Koh et al. (2021) and Sagawa et al. (2020a), we defines 16 overlapping groups by taking the Cartesian product of the binary toxicity label and each of the above 8 demographic identities.
4. **Civilcomments-small:** As Civilcomments is a large dataset of about 448000 datapoints, we create a sub-group stratified subset of 5% for conducting ablations and other detailed experiments. Our subset contains 13770, 2039, and 4866 datapoints in our train, validation, and test split, respectively.

## 4.2 Multitask Model and Training Details

We follow the parameter sharing paradigm (Ruder, 2017; Sener & Koltun, 2018) where both $\mathbf{T}_{\mathrm{end}}$ and $\mathbf{T}_{\mathrm{aux}}$ share the same model body, parameterized by $\theta_{\mathrm{base}}$. We instantiate task-specific heads, parameterized by $\theta_{\mathrm{end}}$ and $\theta_{\mathrm{aux}}$, respectively. We introduce $\ell_1$ regularization to the last layer activations immediately before the per-task prediction heads [*]. Specifically, let $h^{\mathrm{end}}$, $h^{\mathrm{aux}} \in \mathbb{R}^d$ be the output representations generated by the base model, which are fed into their respective task-specific heads. Our final multitask learning objective is expressed as follows:

$$\mathcal{L}_{\mathrm{final}} = \mathcal{L}_{\mathrm{end}} + \alpha_{\mathrm{aux}} \cdot \mathcal{L}_{\mathrm{aux}} + \alpha_{\mathrm{reg}} \left( \|h^{\mathrm{end}}\|_1 + \|h^{\mathrm{aux}}\|_1 \right) \tag{9}$$

We cross-validate optimizing $\mathcal{L}_{\mathrm{final}}$ with different weighting schemes. We choose $\alpha_{\mathrm{aux}}$ and $\alpha_{\mathrm{reg}}$ from the set $\{e^{-1}, e^0, e^1\}$. Note that whilst we optimize $\mathcal{L}_{\mathrm{final}}$ we care only about improving worst-group error on $\mathbf{T}_{\mathrm{aux}}$. We use the pretrained $\mathrm{BERT}_{\mathrm{base}}$ (Devlin et al., 2018) and $\mathrm{ViT}_{\mathrm{base}}$ (Dosovitskiy et al., 2020) as the shared base models for NLP and CV tasks, respectively. We leverage the base models' self-supervised pretraining objectives, namely, masked language modeling (MLM) and masked image modeling (MIM) for our auxiliary transfer task $\mathbf{T}_{\mathrm{aux}}$ as in Dery et al. (2021a). These auxiliary objectives are based on end-task data itself (unless specified otherwise). We do this to maintain an apples-to-apples comparison with our chosen baselines, which do not use external data. As Section 5 will show, we obtain performance improvements even in this setting. In Section 5.4, we show that our improvements when using task-only data are predicated on sufficient prior pre-training. More details on the multitask model and batching scheme are presented in A.1.

---

[*]In contrast to synthetic data experiments, where the norm constraint is applied to pre-prediction layer weights, in this section we directly apply the norm constraint to the features, indirectly constraining model weights. In the synthetic experiment, core and spurious features have a one-to-one mapping to model weights, enabling direct regularization of the features, but that is no longer possible in more complex models.

For training, we vary the fine-tuning learning rate within the ranges of $\{10^{-3}, 10^{-4}\}$ for Waterbirds, and $\{10^{-4}, 10^{-5}\}$ for the text datasets. We experiment with batch sizes in the set $\{4, 8, 16, 32\}$. We use the same batch sizes for $\mathbf{T}_{\text{end}}$ and $\mathbf{T}_{\text{aux}}$. We train for 50 epoch for the NLP datasets and 200 epochs for Waterbirds, with an early stopping patience of 10, as per the check-pointing scheme explained in section 4.2. We use the Adam optimizer for NLP datasets with decoupled weight decay regularization of $10^{-2}$ (Loshchilov & Hutter, 2017). Consistent with the recent studies on ViT (Dosovitskiy et al., 2020; Steiner et al., 2022), we use SGD with a momentum of 0.9 (Sutskever et al., 2013) to fine-tune Waterbirds. We run each hyperparameter configuration across 5 seeds and report the averaged results. We report the ERM, JTT, and groupDRO results for Civilcomments and MultiNLI from Idrissi et al. (2022) as the authors conducted extensive hyperparameter tuning across all these methods. However, since Idrissi et al. (2022) report results on Waterbirds using a ResNet-50 model (He et al., 2016) and our experiments employ ViT, we re-run all baselines using ViT with a consistent set of hyperparameters, as mentioned above.

**Evaluation Details**   We assess all methods and datasets using two model selection strategies:

1. **Val-GP:** This strategy requires group annotations in the validation data during training. Here, we select the model based on the maximum worst-group accuracy on the validation data.
2. **No-GP:** This strategy requires no access to any group annotations during training. We select the model based on the average validation accuracy.

**Baseline Methods**   Since we evaluate our method based on its ability to generalize to worst performing groups, we benchmark it against three popular methods found in group generalization literature. These methods either directly or indirectly optimize for worst-group improvements.

1. **Empirical Risk Minimization (ERM):** This is the standard approach of minimizing the average loss over all the training data. No group information is used during training except when the **Val-GP** strategy is used for model selection.
2. **Just Train Twice (JTT):** It presents a two step approach for worst group generalization Liu et al. (2021). JTT first trains a standard ERM model for $T$ epochs to identify misclassified datapoints. Then, a second model is trained on a reweighted dataset constructed by upweighting the misclassified examples by $\alpha_{up}$. It does not use group information during training except for the **Val-GP** strategy.
3. **Bit-rate Constrained DRO (BR-DRO):** Traditionally, in the two-player formulation of DRO, the adversary can use complex reweighting functions, resulting in overly pessimistic solutions. In contrast, BR-DRO Setlur et al. (2023) constrains the adversary's complexity based on information theory under a data-independent prior. While BR-DRO offers weaker robustness without performance guarantees for arbitrary reweighting, it is less pessimistic and suitable for simpler distribution shifts, characterized by a reweighting function contained in a simpler complexity class. BR-DRO does not use group information during training except for the **Val-GP** setting.
4. **Group-DRO:** Group distributionally robust optimization minimizes the maximum loss across all the sub-groups Sagawa et al. (2020a). This optimization method incorporates group annotations during training. Similar to prior works Liu et al. (2021); Idrissi et al. (2022); Setlur et al. (2023), we treat it as an oracle, as this is the only method that uses group annotations.

## 5   Results And Discussion

In this section, we provide empirical evidence demonstrating the effectiveness of our regularized MTL approach in mitigating worst-group error while maintaining average performance across different scenarios.

### 5.1   Multitasking is Competitive with Bespoke DRO Methods

We first compare our approach with previously proposed methods for tackling worst-group accuracy. Table 3 details the performance of various methods across the tasks of interest for the Val-GP setting. As expected, groupDRO yields the highest worst-group accuracy as it directly optimizes for it. Our MTL approach outperforms JTT and BR-DRO on two datasets (MNLI and Waterbirds) while performing comparatively

Table 3: Mean and standard deviations of the test worst-group accuracies across all the methods under consideration. Regularized MTL consistently reduces the gap between ERM and groupDRO when considering worst-group accuracy.

| Method | Group Labels | Civilcomments | MNLI | Waterbirds |
|--------|--------------|---------------|------|------------|
| ERM | Val Only | $61.3_{2.0}$ | $67.6_{1.2}$ | $85.4_{1.4}$ |
| JTT | Val Only | $67.8_{1.6}$ | $67.5_{1.9}$ | $85.9_{2.5}$ |
| BR-DRO | Val Only | $\mathbf{68.9}_{0.7}$ | $68.5_{0.8}$ | $86.7_{1.3}$ |
| ERM + MT + L1 | Val Only | $68.2_{3.2}$ | $\mathbf{69.7}_{1.5}$ | $\mathbf{87.5}_{2.7}$ |
| groupDRO (Upper Bound) | Train and Val | $69.9_{1.2}$ | $78.0_{0.7}$ | $93.9_{0.7}$ |

with BR-DRO on the CivilComments dataset. Given the competitive results in Table 3, we argue that our regularized MTL formulation is an attractive option over JTT and BR-DRO. Multitasking already features prominently in many ML code bases. Thus, introducing our simple regularization modification to existing MTL implementations represents a smaller technical overhead compared to introducing JTT or BR-DRO to target worst-group error. Also, as we will see in Section 5.2 below, regularized MTL is a single approach capable of improving both worst-group and average accuracy.

## 5.2 Multitasking Improves both Average and worst-group Performance Even in the Absence Group Annotations

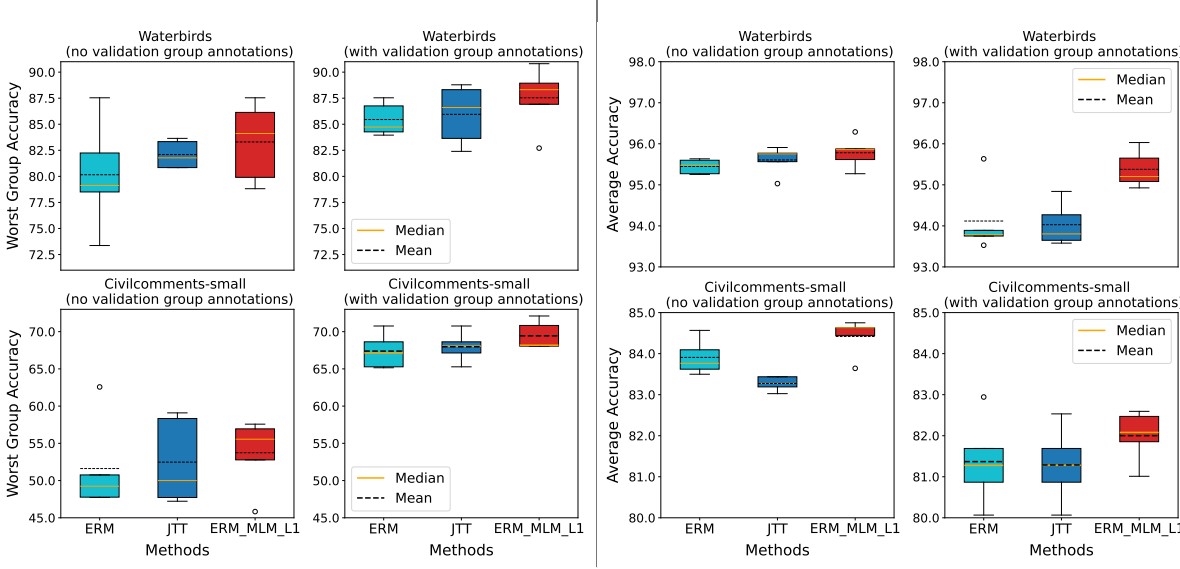

Figure 6: Comparison of the performance of different approaches with respect to average and worst-group accuracy on the Waterbirds dataset under val-GP and no-GP settings. Regularized MTL improves both average and worst-group accuracy even without group annotations. All methods enjoy lift in when validation group annotations are available.

Though previous works typically assume that practitioners have access to the group annotations on the validation set (Liu et al., 2021; Kirichenko et al., 2022), we are interested in settings where no such annotations are available. This covers many tasks of practical interest since, in some cases, it may be prohibitively cost-intensive (financially and in terms of human labor) to acquire group annotations even for the smaller validation set (Paranjape et al., 2023). Consequently, we present a comparative performance analysis in Figure 6, encompassing settings with and without access to group annotations. With respect to worst-group accuracy, our regularized MTL approach outperforms JTT and achieves $\approx 2\%$ lift over ERM when group

annotations are absent, a trend consistent across both Waterbirds and Civilcomments-small datasets. While this lift of $\approx 2\%$ remains when validation group annotations are introduced, the benefit from group-labeled data is more pronounced ($\approx 5\% - 15\%$). This boost can be worthwhile to practitioners who have the resources to obtain some group annotations. Moreover, it becomes evident from Figure 6 that our method not only yields superior worst-group performance but also improves in average performance.

### 5.3 Are both Regularization and Multitasking Jointly necessary?

Table 4: Disentangling the impact of L1 regularization and SSL objective on wost-group accuracy. We see that regularized multitasking is necessary for gains in both average and worst-group performance.

| Dataset | Method | No Group Annotations | | Val Group Annotations | |
|---|---|---|---|---|---|
| | | Avg Acc | WG Acc | Avg Acc | WG Acc |
| Waterbirds | JTT | $95.6_{0.3}$ | $82.1_{1.2}$ | $94.0_{0.5}$ | $85.9_{2.5}$ |
| | ERM | $95.5_{0.2}$ | $80.1_{4.6}$ | $94.1_{0.7}$ | $85.4_{1.4}$ |
| | + L1 | $95.6_{0.3}$ | $82.0_{5.4}$ | $94.7_{0.9}$ | $86.4_{1.4}$ |
| | + MIM | $95.3_{0.4}$ | $80.1_{4.6}$ | $95.0_{0.6}$ | $85.3_{2.4}$ |
| | + MIM + L1 | $\mathbf{95.8}_{0.3}$ | $\mathbf{83.3}_{3.4}$ | $\mathbf{95.4}_{0.4}$ | $\mathbf{87.5}_{2.7}$ |
| Civilcomments-Small | JTT | $83.3_{0.2}$ | $52.5_{5.2}$ | $81.3_{0.8}$ | $68.0_{1.8}$ |
| | ERM | $83.9_{0.4}$ | $51.6_{5.6}$ | $81.4_{1.0}$ | $67.4_{2.1}$ |
| | + L1 | $83.7_{0.4}$ | $51.6_{4.0}$ | $80.3_{0.7}$ | $66.3_{1.6}$ |
| | + MLM | $83.9_{1.2}$ | $\mathbf{58.3}_{6.6}$ | $80.3_{0.7}$ | $68.5_{0.4}$ |
| | + MLM + L1 | $\mathbf{84.4}_{0.4}$ | $53.7_{4.3}$ | $\mathbf{82.0}_{0.5}$ | $\mathbf{69.4}_{1.7}$ |

In this section, we conduct an ablation to verify if ***both*** multitask learning and regularizing the final layer joint embedding space are necessary to improve average and worst-group performance. Our results are captured in Table 4. When assessing the worst-group accuracy, we find that regularizing the final embedding space during ERM can, at times, result in worse performance compared to training via standard ERM ($66.3_{1.6}$ vs $67.4_{2.1}$ on CivilComments-small with validation group labels). On the other hand, multitasking without regularization can fail to improve over ERM, as evinced by the lack of improvement on Waterbirds. The regularized MTL approach is the only setting consistently improving on both datasets with and without validation group annotations. In line with these findings, we observe that joint L1 regularization and multitask learning setup yields the highest average accuracy in most cases.

### 5.4 Impact of Pre-Training

Table 5: Waterbirds: Impact of pre-training on average and worst-group accuracy.

| Pretrianed | Method | No Group Annotations | | Val Group Annotations | |
|---|---|---|---|---|---|
| | | Avg Acc | WG Acc | Avg Acc | WG Acc |
| No | ERM | $65.1_{0.5}$ | $4.5_{1.6}$ | $53.3_{0.7}$ | $10.1_{2.9}$ |
| | JTT | $67.0_{5.3}$ | $\mathbf{10.8}_{12.2}$ | $\mathbf{56.2}_{2.1}$ | $\mathbf{49.9}_{4.0}$ |
| | ERM + MIM + L1 | $\mathbf{67.0}_{2.3}$ | $1.65_{0.7}$ | $53.5_{2.7}$ | $12.0_{3.2}$ |
| yes | ERM | $95.5_{0.2}$ | $80.1_{4.6}$ | $94.1_{0.7}$ | $85.4_{1.4}$ |
| | JTT | $95.6_{0.3}$ | $82.1_{1.2}$ | $94.0_{0.5}$ | $85.9_{2.5}$ |
| | ERM + MIM + L1 | $\mathbf{95.8}_{0.3}$ | $\mathbf{83.3}_{3.4}$ | $\mathbf{95.4}_{0.4}$ | $\mathbf{87.5}_{2.7}$ |

Fine-tuning pre-trained models is arguably the de-facto paradigm in machine learning (Devlin et al., 2018; Dosovitskiy et al., 2020; Dery et al., 2021b). Consequently, our experiments so far have exclusively focused on pre-trained models. In this section, we wish to understand the effect of deviating from this paradigm on our MTL approach. Thus, we compare against JTT and ERM when the model is trained from scratch instead of starting with a pre-trained model. Tables 5 and 6 depict our results on Waterbirds and Civilcomments-small, respectively. Our results show that pre-training is critical for setting up regularized MTL as a viable

Table 6: Civilcomments-small : Impact of pre-training on average and worst-group accuracy.

| Pretrianed | Method | No Group Annotations | | Val Group Annotations | |
|---|---|---|---|---|---|
| | | Avg Acc | WG Acc | Avg Acc | WG Acc |
| No | ERM | $80.7_{0.8}$ | $31.1_{7.2}$ | $\mathbf{74.4}_{0.9}$ | $54.0_{3.7}$ |
| | JTT | $79.6_{0.6}$ | $\mathbf{34.9}_{8.9}$ | $74.3_{1.2}$ | $\mathbf{58.7}_{1.3}$ |
| | ERM+MLM+L1 | $\mathbf{80.7}_{0.6}$ | $31.3_{7.6}$ | $74.2_{0.3}$ | $56.2_{0.9}$ |
| Yes | ERM | $83.9_{0.4}$ | $51.6_{5.6}$ | $81.4_{1.0}$ | $67.4_{2.1}$ |
| | JTT | $83.3_{0.2}$ | $52.5_{5.2}$ | $81.3_{0.8}$ | $68.0_{1.8}$ |
| | ERM+MLM+L1 | $\mathbf{84.4}_{0.4}$ | $\mathbf{53.7}_{4.3}$ | $\mathbf{82.0}_{0.6}$ | $\mathbf{69.4}_{1.7}$ |

remedy against poor worst-group outcomes. We posit the following explanation for this outcome. Note that our informal motivation in Section 2 presupposes an ability to solve the auxiliary task to a reasonable degree. Solving the MLM and MIM tasks effectively from scratch with only the inputs of the relatively small supervised dataset is difficult. This poor performance on the auxiliary task translates to an inability to constrain the use of the spurious features on the end-task. Consistent with prior works (Tu et al., 2020; Wiles et al., 2022), our recommendation to practitioners is to use our approach during the fine-tuning of pre-trained models to be maximally effective.

Another consequence of our findings is that caution is warranted in interpreting the results of previous work on DRO in light of the new paradigm of mostly using pre-trained models. Most previous results on DRO have examined the setting of training from scratch, and as Tables 6 and 5, DRO methods significantly outperform competitors in that setting. However, the originally outsized gains in worst-group error significantly shrink when we move to pre-trained models whilst our method shows superior performance.

## 6  Related Work

- **Multitask Learning.** Multitask learning is a common stratergy for ML practitioners to improve the average performance of their models (Ruder, 2017; Ruder et al., 2019; Liu et al., 2019). While work like Hendrycks et al. (2019; 2020); Mao et al. (2020) have shown that multitasking can improve the adversarial and out-of-distribution robustness of models, the impact of multitasking on worst-group outcomes has been relatively unexplored. Makino et al. (2022) propose generative multitask learning (GMTL), a method that bolsters robustness to target shift by conditioning the input on all available targets, thereby addressing challenges associated with target-causing confounders and spurious dependencies between input and targets. However, it is important to note that their approach necessitates all target annotations during training, a requirement we do not assume in our scenario. Our work is inspired by Gururangan et al. (2020), who introduce constructing auxiliary objectives directly from end-task data for continued pre-training (they dub this Task Adaptive Pre-training – TAPT). Following Dery et al. (2021b; 2023), we multitask this auxiliary task with the end-task. However, unlike these studies, our focus is on improving the worst-case group accuracy of the final model. The work by Michel et al. (2021) explores the balancing of worst and average performance in multitask learning. In their study, they focus on a set of equally important end tasks, striving for proficient model performance across all of them. In contrast, our work delves into the asymmetrical multitask setting, where the presence of the auxiliary task is determined by its contribution to enhancing our target metric in the end task.
- **Robustness using group demographics.** Our multitask learning approach is primarily designed for settings with limited-to-no group annotations. However, many DRO approaches assume the presence of group annotations for all training points. Among the approaches that leverage group information, *Group Distributionally Robust Optimization* Sagawa et al. (2020a) is the most popular technique that tries to minimize the maximum loss over the sub-groups. Goel et al. (2021) presented *Model Patching*, a data augmentation method designed to enhance the representation of minority groups. *FISH*, proposed by Shi et al. (2022), focuses on domain generalization via inter-domain gradient matching. In the settings where group annotations are expensive (financially or in terms of human resources) to procure, these methods are not viable options.

- **Robustness without group demographics.** Extensive research has been dedicated to addressing the challenges of worst-group generalization in the more realistic scenario where access to group annotations during training is unavailable. *GEORGE* Sohoni et al. (2020) adopts a clustering-based methodology to unveil latent groups within the dataset and subsequently employs groupDRO for improved robustness. *Learning from Failure (LfF)* Nam et al. (2020) introduces a two-stage strategy. In the first stage, an intentionally biased model aims to identify minority instances where spurious correlations do not apply. In the second stage, the identified examples are given increased weight during the training of a second model. *Just Train Twice (JTT)* method Liu et al. (2021) follows a similar principle by training a model that minimizes loss over a reweighted dataset. This dataset is constructed by up-weighting training examples misclassified during the initial few epochs. Our regularized MTL approach has several advantages over these methods, even though they are all deployed in the same limited-to-no group annotations settings. As we have demonstrated, our approach can improve both worst-group and average performance, unlike the other approaches targeted against worst-group error only. Secondly, due to the widespread usage of multitask learning by many ML practitioners, implementing our modification represents minimal overhead instead of introducing one of the above bespoke approaches.

## 7 Conclusion

In this work, we presented an empirical investigation of the impact of multitasking on worst-group outcomes. We found that deploying multitasking, *as is*, does not consistently improve upon worst performing groups. We have shown that while DRO methods, like JTT, display superior performance when models are trained from scratch, this is not the case in the currently more widespread setting of fine-tuning a pre-trained model. Specifically, when fine-tuning, our method – regularized multitasking of the end-task with the pre-training objective constructed over end-task data – leads to improvements in worst-case group accuracy over JTT. Our work has demonstrated that it is possible to design a single, simple method that improves both worst-case group accuracy  average accuracy regardless of the availability of group annotations. Since multitask learning is already a standard part of many practitioner's toolbox, and our modification to adapt it against worst-group accuracy is simple, our approach requires minimal overhead to integrate into existing systems compared to bespoke DRO approaches. We thus encourage practitioners to introduce our modification to their MTL pipelines as an essentially free way of improving worst-group performance without sacrificing gains in average performance.

In order to keep an apples-to-apples comparison with DRO approaches, we have primarily focused on multitasking with auxiliary objectives based on end-task data only. For future work, it would be interesting to explore the impact of multitasking with auxiliary objectives based on external data more deeply. It would also be interesting to leverage meta-learning to dynamically adapt the auxiliary tasks towards improving worst-case group outcomes (Dery et al., 2021b; 2023). Additionally, we leave the study on the generalizability of our method to adversarial robustness, domain shift, and label shift to future work.

## 8 Acknowledgements

This work was also supported in part by the Tang AI Innovation Fund, Defence Science and Technology Agency Singapore, National Science Foundation grants IIS1705121, IIS1838017, IIS2046613, IIS2112471, and funding from Meta, Morgan Stanley, Amazon, Google, Schmidt Early Career Fellowship and Apple. Any opinions, findings, conclusions, or recommendations expressed in this material are those of the author(s) and do not necessarily reflect the views of any of these funding agencies.

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

# A  Appendix

## A.1  Training Details

For $\mathbf{T}_{\mathrm{prim}}$, we employ a task-specific classification head of a single-layer multi-layer perceptron (MLP). For $\mathbf{T}_{\mathrm{aux}}$, we leverage the pre-trained MLM and MIM heads from BERT and ViT, respectively. We utilize the embedding of the [CLS] token from the base model through this MLP for classification. To facilitate effective multitask training, we adopt a task-heterogeneous batching scheme Aghajanyan et al. (2021). This facilitates the accumulation of gradients across tasks prior to each parameter update, contributing to improved training efficiency and convergence. Lastly, to ensure proper scaling, the L1 loss is normalized by the number of parameters in the shared representation.

## A.2  Going Beyond the Pre-training Objective

Table 7: Impact of different pre-training objectives on average and worst-group accuracy. Not all auxiliary tasks can help improve worst-group performance.

| Dataset | Method | No Group Annotations | | Val Group Annotations | |
|---------|--------|:-------:|:------:|:-------:|:------:|
| | | Avg Acc | WG Acc | Avg Acc | WG Acc |
| Waterbirds | ERM | $95.5_{0.2}$ | $80.1_{4.6}$ | $94.1_{0.7}$ | $85.4_{1.4}$ |
| | + MIM + L1 | $95.8_{0.3}$ | $83.3_{3.4}$ | $95.4_{0.4}$ | $\mathbf{87.5}_{2.7}$ |
| | + SimCLR + L1 | $\mathbf{96.1}_{0.3}$ | $\mathbf{84.0}_{3.4}$ | $\mathbf{95.5}_{0.7}$ | $87.2_{1.6}$ |
| Civilcomments-Small | ERM | $83.9_{0.4}$ | $51.6_{5.6}$ | $81.4_{1.0}$ | $67.4_{2.1}$ |
| | + MLM + L1 | $\mathbf{84.4}_{0.4}$ | $\mathbf{53.7}_{4.3}$ | $\mathbf{82.0}_{0.5}$ | $\mathbf{69.4}_{1.7}$ |
| | + CLM + L1 | $83.3_{0.7}$ | $50.9_{4.9}$ | $81.1_{0.9}$ | $67.3_{1.4}$ |

Previous works in multitasking with self-supervised objectives suggest that different auxiliary objectives have disparate impacts on end task performance (Dery et al., 2023). Out of curiosity about the impact of the choice of auxiliary objective, we explore the impact of going beyond the model's original pre-training objective. For the Waterbirds dataset, we experiment with SimCLR – a constrastive prediction task based on determining whether two distinct augmented images originate from the same base image (Chen et al., 2020). For BERT experiments on Civilcomments-small, we substitute the standard masked language modeling (MLM) task with causal language modeling (CLM) as the auxiliary task. From the results in Table 7, we observe that SimCLR's performance closely resembles that of the MIM pre-training objective, whereas CLM shows relatively inferior results compared to MLM. We hypothesize that BERT's intrinsic bidirectional attention mechanism and non-autoregressive nature are not ideally suited for causal language modeling (Song et al., 2019), resulting in the model underperforming in our multitask setup. Given the model performance's sensitivity to the replacement objective's choice, we proffer a practical recommendation to practitioners: use the pre-training objective as the auxiliary task. This aligns with recent work on best practices for fine-tuning pre-trained models (Goyal et al., 2023).

## A.3  Bayes optimal model for dimension-independent reconstruction under noised inputs

$$\ell(w_i) = \frac{1}{2}\mathbb{E}\left[(x_i - w_i\tilde{x}_i)^2\right]$$
$$= \frac{1}{2}\mathbb{E}\left[(x_i)^2 - 2w_ix_i\tilde{x}_i + (w_i\tilde{x}_i)^2\right]$$
$$\frac{\partial\ell(w_i)}{\partial w_i} = -\mathbb{E}\left[x_i\tilde{x}_i\right] + w_i\mathbb{E}\left[(\tilde{x}_i)^2\right]$$

The optimal weighting $w_i^*$ is achieved when $\frac{\partial \ell(w_i)}{\partial w_i} = 0$.

$$
\begin{aligned}
w_i^* &= \frac{\mathbb{E}\left[x_i \tilde{x}_i\right]}{\mathbb{E}\left[(\tilde{x}_i)^2\right]} \\
&= \frac{\mathbb{E}\left[x_i(x_i + \epsilon_i)\right]}{\mathbb{E}\left[(x_i + \epsilon_i)^2\right]} \\
&= \frac{\mathbb{E}\left[(x_i)^2\right] + \mathbb{E}\left[x_i\right]\mathbb{E}\left[\epsilon_i\right]}{\mathbb{E}\left[(x_i)^2\right] + 2\mathbb{E}\left[x_i\right]\mathbb{E}\left[\epsilon_i\right] + \mathbb{E}\left[(\epsilon_i)^2\right]} \\
&\quad \text{note } \mathbb{E}\left[\epsilon_i\right] = 0 \ , \ \mathbb{E}\left[(x_i)^2\right] = \sigma_i^2 + 0.5\left(\mu_{i|y=1}^2 + \mu_{i|y=-1}^2\right) \\
&= \frac{\sigma_i^2 + 0.5\left(\mu_{i|y=1}^2 + \mu_{i|y=-1}^2\right)}{\sigma_i^2 + 0.5\left(\mu_{i|y=1}^2 + \mu_{i|y=-1}^2\right) + \sigma_{\text{noise}}^2}
\end{aligned}
$$

## B  Broader Impact Statement

In terms of broader impact, our work has beneficial implications for group fairness and mitigating demographic-based bias in machine learning systems. Specifically, we present a simple approach to improving worst-case group error. This means that our work contributes positively to certain groups that would otherwise be impacted by the poor performance of ML models. We have also provided a new lens from which to view the problem of the impacts of multitasking, resulting in showing that it is tool able against poor group-based outcomes. However, our method introduces compute overhead due to the optimization of multiple objectives. This results in increased power consumption and, thus, greenhouse emissions during the training of models. Nevertheless, introducing our simple regularization modification to existing MTL implementations represents a negligible technical overhead than introducing JTT or BR-DRO to target worst-group error.

Table 8: Sensitivity analysis of all the methods to different hyperparameters wrt worst-group accuracy.

| Dataset | Method | Seed | Learning Rate | Batch Size | Up | T | $\lambda_{aux}$ | $\lambda_{reg}$ |
|---|---|---|---|---|---|---|---|---|
| Waterbirds | ERM | 0.0734 | 0.5717 | $-0.4384$ | – | – | – | – |
| | JTT | 0.0573 | – | $-0.380$ | 0.0246 | 0.1862 | – | – |
| | ERM + MT + L1 | $-0.0314$ | – | 0.0142 | – | – | 0.0943 | 0.4114 |
| | groupDRO | $-.1688$ | 0.196 | $-0.1384$ | – | – | – | – |
| Civilcomments-Small | ERM | 0.0617 | $-0.8$ | 0.1029 | – | – | – | – |
| | JTT | $-0.0315$ | – | 0.2104 | $-0.0047$ | 0.1111 | – | – |
| | ERM + MT + L1 | 0.0677 | – | 0.0453 | – | – | 0.0601 | $-0.2955$ |
| | groupDRO | $-0.0392$ | $-0.7352$ | $-0.1726$ | – | – | – | – |

**Sensitivity to Hyperparameters:** For evaluating the sensitivity of worst group accuracy to various hyperparameters across different methods, we employ Kendall's rank coefficient $\tau$ Kendall (1938). To adhere to our computation budget, we opt for the optimal learning rate utilized in ERM for both JTT and ERM+MT+L1. Our sensitivity analysis from Table 8 reveals distinct preferences in learning rates. The ViT model applied to waterbirds, demonstrating a preference for higher learning rates. Conversely, when trained on the civilomments-small dataset, BERT exhibits a preference for lower learning rates. The epoch at upsampling is also a sensitive hyperparameter for JTT. The sensitivity to different seed values is negligible across all the methods. Notably, our method displays the most minor sensitivity to changes in batch size. However, it is noteworthy that our method is most responsive to variations in the regularization parameter ($\lambda_{reg}$), with lower values resulting in higher worst group accuracy.

