# OpenReview forum: "Multitask Learning Can Improve Worst-Group Outcomes"
_TMLR — Accepted by TMLR_

### Review · Reviewer_DGj8 · 2023-12-29

**Summary Of Contributions:**

The paper investigates how multitask learning (MTL) affects worst-group outcomes, which are the performance of a model on the most disadvantaged subgroups of data.

The paper proposes regularized MTL, which combines the end task with a pre-training objective and applies ℓ1 regularization to the shared representation space, as a simple and effective method to improve worst-group outcomes.

The paper provides both synthetic and natural data experiments to show that regularized MTL can induce the model to rely more on the core features and achieve better worst-group accuracy than distributionally robust optimization (DRO) methods.

**Audience:**

Yes

**Broader Impact Concerns:**

The paper does not have a Broader Impact Statement section. The paper should add a Broader Impact Statement section that discusses the potential ethical, social, or environmental implications of the work, both positive and negative.

Some possible broader impact concerns of the work are:

The paper could have a positive impact on improving safety and group fairness and reducing discrimination in machine learning applications, especially for those that use pre-trained models and MTL. The paper could also inspire more research on understanding and mitigating the effects of spurious correlations and feature selection on group outcomes.

**Claims And Evidence:**

Yes

**Requested Changes:**

The paper should be restructured and define the problem first and then introduce the method, results and supporting evidence. It should improve the readability and clarity of some sections, such as the problem formulation, the synthetic data experiments, and the appendix. For example, the paper could define the notation and the terminology more explicitly, and provide more explanations and examples for the synthetic data experiments. The paper could also reorganize the appendix and provide more details and proofs for some of the claims and results.

The paper should mention the contribution and novelty of the work explicitly in the introduction.
The paper should compare the proposed method with other MTL methods that use different auxiliary tasks or different regularization techniques. This would help to understand the advantages and disadvantages of the proposed method, and how it relates to existing MTL literature.

**Strengths And Weaknesses:**

Strengths:

The paper addresses an important and timely problem of group fairness in machine learning, especially in the context of pre-trained models and MTL.

The paper demonstrates the effectiveness of the proposed method on three natural datasets across computer vision and natural language processing tasks and shows that it consistently outperforms several state-of-the-art DRO methods on both worst and average group outcomes, regardless of the availability of group annotations.

The paper did several ablation experiments on the effect of their method and pertaining.

Weaknesses:

The paper lacks a structure, as of this version, the introduction consists of methods and results as well.

The paper does not provide a clear and rigorous definition of the problem and the proposed method.

The paper does not provide a clear explanation of why pre-training is critical for the success of the proposed method, and how it affects the trade-off between core and spurious features.

The paper does not compare the proposed method with other MTL methods that use different auxiliary tasks or different regularization techniques, which could provide more insights into the effectiveness of the proposed method.

The paper needs to address the potential limitations or drawbacks of the proposed method, such as the computational cost, the sensitivity to pre-task, hyperparameters, or the generalization to other domains or tasks.

---

> ### Author Response · Authors · 2024-01-16
> **Response to reviewer DGj8**
>
> We appreciate the reviewer's insightful comments and valuable suggestions. In our response below, we strive to address each query comprehensively.
>
> 1. *The paper does not provide a clear and rigorous definition of the problem and the proposed method.*
>
> Thanks for the suggestion. We have included a problem definition section in the paper for more clarity.
>
> 2. *The paper lacks a structure, as of this version, the introduction consists of methods and results as well.  / The paper should be restructured and define the problem first and then introduce the method, results and supporting evidence. It should improve the readability and clarity of some sections, such as the problem formulation, the synthetic data experiments, and the appendix. For example, the paper could define the notation and the terminology more explicitly, and provide more explanations and examples for the synthetic data experiments. The paper could also reorganize the appendix and provide more details and proofs for some of the claims and results.*
>
> Thank you for your suggestion. We have made revisions to include an explicit problem statement section. Whilst it is conventional to have the structure that the reviewer kindly described, we found that the eventual structure we use in the paper was the best way to convey our ideas in the simplest way possible whilst catering to a wide audience. Furthermore, we would like to acknowledge that reviewer wKJf  points out that our “narrative is skillfully presented”.
>
> 3. *The paper does not provide a clear explanation of why pre-training is critical for the success of the proposed method, and how it affects the trade-off between core and spurious features*
>
> Please note that we do discuss the reason for pre-training being critical in section 5.4. We mention that **Note that our informal motivation in Section 2 presupposes an ability to solve the auxiliary task to a reasonable degree. Solving the MLM and MIM tasks effectively from scratch with only the inputs of the relatively small supervised dataset is difficult**.  The auxiliary objective is only useful if we are able to solve it to a reasonable degree. This is not easy when we are training the model from scratch.
>
> 4. *The paper does not compare the proposed method with other MTL methods that use different auxiliary tasks or different regularization techniques, which could provide more insights into the effectiveness of the proposed method.*
>
> Comparison with other MTL methods: To the best of our knowledge,  ours is the first work investigating the impact of multitask learning on group fairness. Whilst there are other methods for optimizing over MTL objectives, we prioritized simplicity by sticking to the dominant shared-body, different heads with weighted objectives approach. In terms of different choices in auxiliary tasks,  note that in Section A.2, we try out MTL with different auxiliary objectives and give recommendations about which objectives to use.
>
> 5. *The paper needs to address the potential limitations or drawbacks of the proposed method, such as the computational cost, the sensitivity to pre-task, hyperparameters, or the generalization to other domains or tasks.*
> The reviewer's suggestion regarding the absence of limitations or drawbacks section is duly noted. We have added a broader impact statement (in the main paper) and have also included sensitivity analysis in the updated appendix.

---

> > ### Author Response · Authors · 2024-01-20
> > **Reminder to evaluate the rebuttal**
> >
> > Dear Reviewer,
> >
> > As the rebuttal period is coming to an end (the final deadline is 22nd Jan 2024), we are eager to know whether we have successfully addressed your concerns. If we have, we would greatly appreciate it if you would consider revising your recommendation accordingly. Should you have any further questions, we are more than willing to provide additional clarifications.
> >
> > Best regards,
> >
> > Authors

---

### Review · Reviewer_wKjf · 2024-01-07

**Summary Of Contributions:**

The authors propose that multitask learning applied to a discriminative task and an input-reconstruction task can improve worst-case and average-case accuracy for the discriminative task. There exist two types of features: core and spurious. Solely training on the discriminative task can result in the core features being ignored. When the input-reconstruction task is simultaneously optimized with strong regularization, this ensures the core features are also learned, thus improving robustness.

**Audience:**

Yes

**Claims And Evidence:**

Yes

**Requested Changes:**

I would add some discussion about the choice of regularization, and why you went with L1 norm.

I also suggest discussing https://arxiv.org/abs/2202.04136 in the MTL section of the related work. It is about improving robustness to shifts in p(y, y') in multitask learning, where y and y' are the targets. I believe it's related to this work, since yours is about improving robustness to shifts in p(y, s), where s is the spurious attribute in Section 2. Both can be framed as improving robustness to unobserved confounding that is particular to MTL.

**Strengths And Weaknesses:**

This is a strong paper that delivers a simple but meaningful message in a very clear way. Toy problems are used skillfully to present a narrative that is easy to follow, and left me believing that the method should work, even before seeing the empirical results. The empirical results are also strong, and support the authors' claims. I appreciated the level of detail the authors devoted in their discussion of model selection, which I believe is crucial for a paper about about robustness.

The only weakness I found is the authors' lack of discussion on their choice of L1 norm regularization (as opposed to other forms, such as L2 norm or dropout). The choice of regularization seems important given how critical it is to their proposed approach.

---

> ### Author Response · Authors · 2024-01-16
> **Response to reviewer wKjf**
>
> We appreciate the reviewer's insightful comments and valuable suggestions. In our response below, we strive to address each query comprehensively.
>
> 1. *The only weakness I found is the authors' lack of discussion on their choice of L1 norm regularization (as opposed to other forms, such as L2 norm or dropout). The choice of regularization seems important given how critical it is to their proposed approach.*
>
> Why use the L1 norm? As mentioned in Section 4.3, we would like to clarify that we do not treat L1 as a conventional regularizer; instead, we leverage it as a method to induce sparsity in the features, thus creating competition between core and spurious features. Note that in principle, we would like to drive the spurious features to 0 since any non-zero value could still create opportunities for the model to latch on to the spurious features. L1 constraint satisfies this need, whilst the other options  – L2 and dropout do not.  Whilst L1 is the principled choice, please also find results from experiments we run to compare against L2 regularization. We conduct an experiment on the Waterbirds dataset to validate that our choice of L1 norm is indeed superior. **We achieve a worst-group accuracy of 84.48 ± 3.54, when we switch to L2 norm, similar to the ERM results. Our L1 norm result however was 87.5 ± 2.7.  This highlights the need for L1 regularization over L2 in our setup**.
>
> 2. *I also suggest discussing https://arxiv.org/abs/2202.04136 in the MTL section of the related work. It is about improving robustness to shifts in p(y, y') in multitask learning, where y and y' are the targets. I believe it's related to this work, since yours is about improving robustness to shifts in p(y, s), where s is the spurious attribute in Section 2. Both can be framed as improving robustness to unobserved confounding that is particular to MTL.*
>
> We thank the reviewer for suggesting this work. The paper is very relevant to our setting. We have updated the related work to incorporate a discussion of this work:  Makino et al. (2022) propose generative multitask learning (GMTL), a method that bolsters robustness to target shift by conditioning the input on all targets, thereby addressing challenges associated with target-causing confounders and spurious dependencies between input and targets. However, it is important to note that their approach necessitates all target annotations during training, a requirement that we do not assume in our scenario.

---

> > ### Author Response · Authors · 2024-01-20
> > **Reminder to evaluate the rebuttal**
> >
> > Dear Reviewer,
> >
> > As the rebuttal period is coming to an end (the final deadline is 22nd Jan 2024), we are eager to know whether we have successfully addressed your concerns. If we have, we would greatly appreciate it if you would consider revising your recommendation accordingly. Should you have any further questions, we are more than willing to provide additional clarifications.
> >
> > Best regards,
> >
> > Authors

---

### Review · Reviewer_tsay · 2024-01-08

**Summary Of Contributions:**

The paper proposes to use masked image/language modeling as an auxiliary task to improve the worse-group performance in datasets with subgroup shifts. On a synthetic dataset, the auxiliary task is shown to be quite effective when the model is trained with a proper regularizer. On 4 real-world datasets, the method achieves better worse-group accuracy than JTT and BR-DRO when there is no group labels.

**Audience:**

Yes

**Claims And Evidence:**

Yes

**Requested Changes:**

Elaborate the gap between the model complexity measure in synthetic and real-world data, and provide some thoughts on why the feature regularization is so important for MLM to work well.

Test if the method is scalable to large-scale vision datasets.

**Strengths And Weaknesses:**

Strengths:

1. The paper is the first to demonstrate that the subgroup shift robustness can be improved by masked image/language modeling as the auxiliary task, as far as I know.

2. In the synthetic data experiment, the contrast between Fig. 4 and Fig. 5 looks compelling.


Weaknesses:

1. Section 3.3 uses the weight norm as the measure for model complexity, while Section 4.2 uses the last layer feature norm, which means there is a gap between the argument in the synthetic data and in the real data. Resolving the inconsistency is important to understand the effect the the regularization term, which is quite essential as Fig. 5 shows.

2. The real data experiment on image data is only done on a small synthetic dataset, it is unclear whether the method will scale to larger datasets like WILDS-iWildCam [2].

3. Some papers on multi-task learning for out-of-distribution generalization might be relevant to the paper, e.g., [1].


[1] Albuquerque, Isabela, et al. "Improving out-of-distribution generalization via multi-task self-supervised pretraining." arXiv preprint arXiv:2003.13525 (2020).

[2] https://wilds.stanford.edu/datasets/

---

> ### Author Response · Authors · 2024-01-16
> **Response to Reviewer tsay**
>
> We appreciate the reviewer's insightful comments and valuable suggestions. In our response below, we strive to address each query comprehensively.
>
> 1. *Section 3.3 uses the weight norm as the measure for model complexity, while Section 4.2 uses the last layer feature norm, which means there is a gap between the argument in the synthetic data and in the real data. Resolving the inconsistency is important to understand the effect of the regularization term, which is quite essential as Fig. 5 shows*
>
> We agree with the reviewer's insight into the identified gap and appreciate their constructive feedback. Our synthetic experiments provide a simplified picture that motivates our real-world setting experiments. In case of our experiments with more complicated models, we found that it is easiest to apply the norm constraint directly to the features instead of the weights, since this indirectly also constrains the model weights themselves. Note that in the synthetic experiment, the core and spurious features are scalars with 1-1 mapping to model weights. This makes it easy to “setup competition between features” by applying regularization to the model weights directly. When we move from scalar weights to vector based “feature” weights, it becomes unclear how to apply  the weight norm regularization so that the competition is at the level of features.  We have updated the paper to make this clearer and to explain our choice.
>
> 2. *The real data experiment on image data is only done on a small synthetic dataset, it is unclear whether the method will scale to larger datasets like WILDS-iWildCam [2]*
>
> Our research primarily centers around group-fairness rather than domain generalization. The WILDS-iWildCam dataset is designed for domain generalization, and so it tests changes in the support of the input distributions. In contrast, datasets addressing group fairness, such as Civilcomments, MNLI, and Waterbirds, feature data from the same support but encompass different demographic groups. Thus WILDS-iWildCam is not an appropriate dataset for our setting of interest.  In terms of scale, note that our CivilComments-large dataset has ~500k samples and from Table 3, we achieve competitive performance compared to other methods, indicating that our method does scale.

---

> > ### Author Response · Authors · 2024-01-20
> > **Reminder to evaluate the rebuttal**
> >
> > Dear Reviewer,
> >
> > As the rebuttal period is coming to an end (the final deadline is 22nd Jan 2024), we are eager to know whether we have successfully addressed your concerns. If we have, we would greatly appreciate it if you would consider revising your recommendation accordingly. Should you have any further questions, we are more than willing to provide additional clarifications.
> >
> > Best regards,
> >
> > Authors

---

### Decision · Action_Editor_JUkP · 2024-02-14

**Recommendation:** Accept with minor revision

**Comment:**

This paper aims to improve the worst-case performance of a given model across groups by introducing a masked reconstruction loss. After building intuition in toy settings, the proposed approach is compared to various baselines on standard benchmarks. Reviewers generally found the approach convincing and appreciated the clarity and structure of the paper, though one reviewer made suggestions about the presentation that should be considered. Otherwise, the various reviewers' concerns were addressed during the rebuttal.

**Audience:**

Yes. One reviewer noted that "The exploration conducted in the paper offers valuable insights that would be of interest to the TMLR audience".

**Claims And Evidence:**

Yes. Generally the reviewers felt the experiments in the paper supported the claims, and that the inclusion of toy settings to better understand the method was valuable. One reviewer noted that "experimental evidence strongly backs up its claims".